# One-pot catalytic synthesis of urea derivatives from alkyl ammonium carbamates using low concentrations of $CO_2$

Hiroki Koizumi [1], Katsuhiko Takeuchi [1✉], Kazuhiro Matsumoto [1], Norihisa Fukaya[1], Kazuhiko Sato[1], Masahito Uchida[2], Seiji Matsumoto[3], Satoshi Hamura[3] & Jun-Chul Choi [1✉]

To reduce anthropogenic carbon dioxide ($CO_2$) emissions, it is desirable to develop reactions that can efficiently convert low concentrations of $CO_2$, present in exhaust gases and ambient air, into industrially important chemicals, without involving any expensive separation, concentration, compression, and purification processes. Here, we present an efficient method for synthesizing urea derivatives from alkyl ammonium carbamates. The carbamates can be easily obtained from low concentrations of $CO_2$ as present in ambient air or simulated exhaust gas. Reaction of alkyl ammonium carbamates with 1,3-dimethyl-2-imidazolidinone solvent in the presence of a titanium complex catalyst inside a sealed vessel produces urea derivatives in high yields. This reaction is suitable for synthesizing ethylene urea, an industrially important chemical, as well as various cyclic and acyclic urea derivatives. Using this methodology, we also show the synthesis of urea derivatives directly from low concentration of $CO_2$ sources in a one-pot manner.

[1] National Institute of Advanced Industrial Science and Technology (AIST), Tsukuba, Ibaraki, Japan. [2] Tosoh Corporation, Advanced Materials Research Laboratory, Ayase, Kanagawa, Japan. [3] Tosoh Corporation, Minato-ku, Tokyo, Japan. ✉email: takeuchi-k@aist.go.jp; junchul.choi@aist.go.jp

Climate change due to emission of greenhouse gases is one of the most pressing challenges globally. Among the greenhouse gases, carbon dioxide ($CO_2$) emission due to human activities has been considered the main cause of global warming[1]. Therefore, it is desirable to develop technologies for $CO_2$ capture, storage, and utilization (CCSU) from exhaust gases and, if possible, air, for reducing $CO_2$ in the atmosphere[2–6]. The method for capturing $CO_2$ directly from ambient air is called direct air capture (DAC)[7–10]. In the field of classical organic synthesis, $CO_2$ has been extensively utilized as an abundant, low-toxicity, and low-cost C1 building block[11–16]. However, these reactions require highly pure $CO_2$, often under high pressure. In other words, these reactions cannot be realized using low concentrations of $CO_2$ directly and require prior steps, such as concentration, compression, and purification, which involve high costs. Owing to the high costs involved, organic syntheses using $CO_2$ are considered reasonable only for high value-added chemicals, such as pharmaceuticals at CCSU. Thus, new methods that do not require high-$CO_2$ concentrations, compression, or purification must be developed for CCSU to synthesize a wide range of chemical products. The DAC method has been utilized by Yoshida et al. and Inagaki et al. independently for the synthesis of oxazolidinone derivatives from $CO_2$ in air and propargylamine derivatives[17–19]. However, in order to reduce $CO_2$ emissions, core chemicals that are handled in large quantities must be synthesized, using inexpensive raw materials and low concentrations of $CO_2$. In this regard, the abovementioned reactions use expensive raw materials, and the products formed are not core chemicals.

Urea derivatives ($R_2N(CO)NR_2$) are used as solvents, medicines, and fertilizers[20–22]. In particular, ethylene urea, a cyclic urea derivative, is a relatively high value-added core chemical (ca. 10 USD/kg in 2018) with a large market (ca. 12,000 t/year in 2018). It is used in paints and as a fabric finishing agent and raw material for agrochemicals[20,23]. Synthesis of urea derivatives has been performed using $CO_2$ as a raw material. Reactions of amines with high-pressure $CO_2$ in the presence of catalysts, such as cesium hydroxide[24], cerium oxide[25,26], organic bases[27], and inorganic bases[28] have been reported. Recently, the synthesis of urea derivatives from amines and atmospheric $CO_2$, with an equivalent amount of phosphine and trichloroisocyanuric acid as a sacrificial reagent, has also been reported[29]. These studies are, of course, important as they provide methods for utilizing less reactive $CO_2$ as a C1 building block. However, not all such studies contribute to the reduction of $CO_2$ emission, since the reactions require high-purity $CO_2$ either under high-pressure or in the presence of sacrificial reagents.

Considering the above facts, we inferred that the development of a reaction to synthesize urea derivatives from low concentrations of $CO_2$, without the need for $CO_2$ concentration, compression, and purification, could be a useful method for reducing $CO_2$ emissions, while maintaining market competitiveness. Alkyl ammonium carbamates are well-known intermediates in the synthesis of urea derivatives from $CO_2$ and amines[24–29], and are also extensively studied in the field of $CO_2$ capture[2–10,30]. Thus, we focused on utilizing alkyl ammonium carbamates, which are 1:2 adducts of $CO_2$ and amines, as intermediates for the synthesis of urea derivatives from low concentrations of $CO_2$. Since alkyl ammonium carbamates reversibly decompose into $CO_2$ and amine when heated, they are utilized as a mediator for capturing and concentrating $CO_2$ from exhaust gases released from major high-$CO_2$ outlets, such as thermal power plants, cement industries, and gas refineries.

In this study, we have developed an efficient method for the synthesis and isolation of alkyl ammonium carbamates from aliphatic amines and low concentrations of $CO_2$, including $CO_2$ from ambient air. Using these synthesized alkyl ammonium carbamates, we have developed a method for the catalyzed synthesis of various important urea derivatives, such as ethylene urea. Most importantly, this method does not require separate steps for $CO_2$ concentration, compression, and purification and hence, is expected to be an efficient and cost-effective strategy for reducing $CO_2$ emissions.

## Results and discussion

**Synthesis of alkyl ammonium carbamates from low concentrations of $CO_2$.** Inagaki et al. have reported the DAC of $CO_2$ to produce alkyl ammonium carbamates by exposing benzylamine and its analogs in ambient air for 1 week[30]. However, the efficiency of formation and the isolation methods for alkyl ammonium carbamates under various $CO_2$ concentrations have not been discussed. Therefore, we investigated the efficiency of alkyl ammonium carbamate formation as a function of $CO_2$ concentration, using ethylenediamine, the raw material for ethylene urea, as a model compound (Fig. 1). The experiment was conducted by bubbling pure $CO_2$ gas or $CO_2/N_2$ mixed gas (50 or 15 vol% $CO_2$) at a flow rate of 50 mL/min into 50 mL EtOH solution of ethylenediamine (20 mmol). Efficiency of alkyl ammonium carbamate (**1a**) formation was estimated based on the conversion rate of ethylenediamine, which was calculated from the $^1H$ NMR spectra, using mesitylene as an internal standard. When pure $CO_2$ gas was used, 97% ethylenediamine was converted after 10 min of bubbling (Fig. 1, red curve). When $CO_2/N_2$ mixed gas was bubbled, it took 20 min for 50 vol% $CO_2$ (Fig. 1, green curve) and 70 min for 15 vol% $CO_2$ (Fig. 1, blue curve) to consume ~97% ethylenediamine. In other words, although the rate of carbamate formation depended on the $CO_2$ concentration, the carbamate could be obtained in high yields even with a low concentration of $CO_2$ if the reaction time was prolonged. Since **1a**

a)

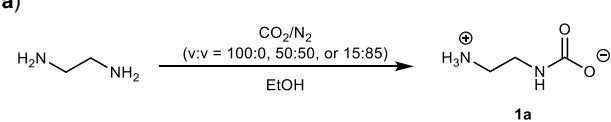

b)

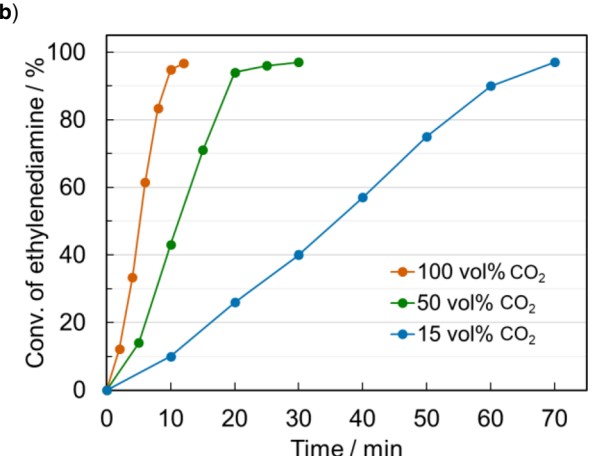

**Fig. 1 Efficiency of formation of 1a by bubbling various concentrations of $CO_2$. a** Synthesis of **1a** from ethylenediamine by bubbling various concentrations of $CO_2$. Reaction conditions: ethylenediamine (1.2 g, 20 mmol), EtOH (50 mL), gas flow rate (50 mL/min). **b** Plot of the conversion rate of ethylenediamine as a function of bubbling time of pure $CO_2$ gas and $CO_2/N_2$ mixed gases: red curve: pure $CO_2$, green curve: $CO_2/N_2 = 50:50$ (v/v), and blue curve: $CO_2/N_2$: 15:85 (v/v). Conversions of ethylenediamine were calculated from $^1H$ NMR spectra using mesitylene as an internal standard.

**Table 1 Synthesis and isolation of 1a from various $CO_2$ sources.**

| Entry | $CO_2$ source | Solvent | Method | Time | Isolated yield (%) |
|---|---|---|---|---|---|
| 1 | Pure $CO_2$ | EtOH | Bubbling | 10 min | 93 |
| 2 | $CO_2/N_2$ (v:v = 15:85) | EtOH | Bubbling | 30 min | 96 |
| 3 | Simulated exhaust gas[a] | EtOH | Bubbling | 30 min | 92 |
| 4 | Ambient air | — | Exposing | 1 week | 45 |

Reaction conditions: ethylenediamine (3.0 g, 50 mmol), solvent (50 mL).
[a]$CO_2$: 15 vol%, $N_2$: 85 vol%, CO: 300 p.p.m., $NO_2$: 500 p.p.m., $SO_2$: 500 p.p.m.

precipitates in EtOH as a white powder after bubbling $CO_2$-containing gases, it could be easily separated by filtration. The isolated yields of **1a** from various $CO_2$ sources are summarized in Table 1. The isolated yields of **1a** from both pure $CO_2$ and 15 vol% $CO_2$ were high (93–96%) if the reaction was prolonged (Table 1, entries 1 and 2). This volume percent of $CO_2$ (15 vol%) is approximately same as the $CO_2$ concentration in exhaust gases of industrial plants, such as thermal power plants, cement plants, petroleum refinery, and steel mills, and $CO_2$ emissions from these sources have raised concerns worldwide. Although the exhaust gases from industrial plants contain various compounds, such as water, CO, $H_2$, $SO_x$, and $NO_x$, we expect most of these impurities to be easily removed by precipitation and filtration in our method for the synthesis of **1a**. In fact, when the same experiment was conducted using simulated exhaust gas ($CO_2$: 15 vol%, $N_2$: 85 vol%, CO: 300 p.p.m., $NO_2$: 500 p.p.m., $SO_2$: 500 p.p.m.; the content of each component was determined according to the recent report by Kuznetsov et al.[31]), the yield of **1a** was almost as high as that obtained when 15% $CO_2$ gas was used (92%, Table 1, entry 3). Therefore, the method for synthesizing **1a** developed in this study should be effective in utilizing $CO_2$ of exhaust gases from industrial plants. Furthermore, since ethylenediamine reacts with $CO_2$ in ambient air (0.04 vol%), **1a** can also be synthesized by DAC. In fact, **1a** could be isolated in 45% yield by exposing ethylenediamine to ambient air for 1 week (Table 1, entry 4). The reaction of amine with $CO_2$ is known to yield water-incorporated ammonium carbonate salts, depending on the reaction conditions[30]. However, elemental analysis showed that anhydrous alkyl ammonium carbamate **1a** could be obtained selectively from each $CO_2$ source by our method, even when undried EtOH was used. In addition, the alkyl ammonium carbamates synthesized using various $CO_2$ source were of comparable quality, as no significant differences were observed in the [1]H and [13]C NMR spectra and elemental analysis (Supplementary Figs. 1–3 and Supplementary Table 1). Since **1a** was only soluble in water, the NMR spectra was acquired in $D_2O$; however, some amount of **1a** reacted with $D_2O$ to form a deuterated ammonium bicarbonate salt (Supplementary Figs. 1–3 for details). Therefore, the purity of **1a** can be confirmed only by elemental analysis.

**Synthesis of urea derivatives from alkyl ammonium carbamates.** Next, we explored the optimal conditions for the catalytic synthesis of ethylene urea **3a** from **1a**. Since alkyl ammonium carbamates easily decompose into amines and $CO_2$ upon heating, we performed the reaction inside a sealed vessel that was completely filled with the substrate, catalyst, and solvent, to prevent the release of $CO_2$ from the reaction system. We investigated Lewis acids, such as group 4 metal complexes and tin compounds. These are effective catalysts for $CO_2$ utilization reactions involving dehydration, such as the synthesis of dialkyl carbonates and

organic carbamates, using acetals as a regenerable dehydration agent[32–34]. In addition to CsOH, $CeO_2$, DBU, and inorganic bases, which are effective for the synthesis of urea derivatives using high-pressure $CO_2$ (refs. [24–28]), the Lewis acid catalysts can be used to synthesize urea derivatives from alkyl ammonium carbamate. As a result, group 4 metal compounds were found to be particularly active (Supplementary Table 2). The most active catalyst was a Ti complex $Cp_2Ti(OTf)_2$ (Cp = cyclopentadienyl; OTf = trifluoromethanesulfonate; Table 2, entry 1). When a 10 mL autoclave having a dead volume was used instead of the sealed reaction vessel, the yield of **3a** decreased (Table 2, entry 2). Almost quantitative **3a** was obtained in 15 h of reaction time (Table 2, entry 3). Optimization of the solvents revealed that the yields were better in 1,3-dimethyl-2-imidazolidinone (DMI) and N-methylpyrrolidone (NMP) than in other polar solvents, such as 2-pyrrolidone, dimethyl formamide, tetrahydrofuran, acetonitrile, and 1,4-dioxane (Table 2, entries 4–9). In order to clarify the dependence of reaction efficiency on the solvents, the reaction was conducted in DMI, NMP, and 1,4-dioxane without any metal catalyst. The reaction proceeded to some extent in the absence of a catalyst when DMI and NMP were used (Table 2, entries 10–12). It was also found that the addition of one equivalent of ethylene urea **3a** accelerated the reaction (Table 2, entry 13).

We believe that the solvent and autocatalytic effects discussed above are closely related to the reaction mechanism. The Ti-catalyzed synthesis of urea derivatives from alkyl ammonium carbamate uses a closed vessel with no dead volume, making it difficult to follow the detail of the reaction. Nevertheless, we assumed this reaction to proceed by a mechanism via a metal carbamate, similar to the $CeO_2$-catalyzed synthesis using high-pressure $CO_2$, where the metal center acts as a Lewis acid, as proposed by Tomishige et al. (Fig. 2)[25]. It should be noted that in the $CeO_2$ catalytic system, dicarbamic acid is formed once due to the presence of a large amount of $CO_2$; however, in the Ti complex catalytic system used in this study, the titanium carbamate complex could be obtained directly because there is no additional $CO_2$ (Fig. 2, step 1). In the $CeO_2$ catalytic system, it is assumed that the oxygen atom in $CeO_2$ acts as a proton acceptor, enabling the nucleophilic attack of the amine and promoting the reaction. On the other hand, in the Ti complex catalytic system, basic solvents such as DMI and NMP assists the proton transfer and acted as temporary proton acceptors, thereby promoting the reaction (Fig. 2, steps 2 and 3). Kim and Lee et al. reported that the addition of ethylene urea stabilized zwitterionic carbamate species via a hydrogen-bonding network during the synthesis of urea derivatives, using inorganic salt catalysts; similar acceleration effects are also expected in our Ti complex catalytic system[28]. Finally, the Ti complex catalyst would regenerate through the dehydration reaction and subsequent formation of urea derivatives (Fig. 2, steps 4 and 5).

**Table 2 Optimization of reaction conditions for the synthesis of ethylene urea 3a from 1a.**

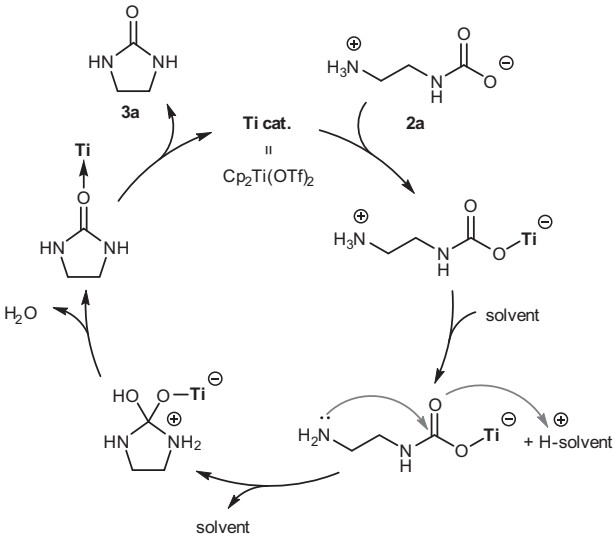

| Entry | Catalyst | Solvent | Time (h) | 3a (%) |
|---|---|---|---|---|
| 1 | Cp$_2$Ti(OTf)$_2$ | DMI | 3 | 52 |
| 2[a] | Cp$_2$Ti(OTf)$_2$ | DMI | 3 | 43 |
| 3 | Cp$_2$Ti(OTf)$_2$ | DMI | 15 | 99 |
| 4 | Cp$_2$Ti(OTf)$_2$ | NMP | 15 | 99 |
| 5 | Cp$_2$Ti(OTf)$_2$ | 2-Pyrrolidone | 15 | 73 |
| 6 | Cp$_2$Ti(OTf)$_2$ | DMF | 15 | 17 |
| 7 | Cp$_2$Ti(OTf)$_2$ | Acetonitrile | 15 | 44 |
| 8 | Cp$_2$Ti(OTf)$_2$ | THF | 15 | 79 |
| 9 | Cp$_2$Ti(OTf)$_2$ | 1,4-Dioxane | 15 | 88 |
| 10 | — | DMI | 15 | 31 |
| 11 | — | NMP | 15 | 32 |
| 12 | — | 1,4-Dioxane | 15 | 9 |
| 13[b] | — | DMI | 15 | 51 |
| 14[c] | Cp$_2$Ti(OTf)$_2$ | DMI | 24 | 99 (82[d]) |

Reaction conditions: **1a** (2.0 mmol), catalyst (0.20 mmol), solvent (4.5 mL), 5 mL sealed reaction vessel. Yields of **3a** were determined by [1]H NMR using mesitylene as an internal standard.
*Cp* cyclopentadienyl, *OTf* trifluoromethanesulfonate, *DMI* 1,3-dimethyl-2-imidazolidinone, *NMP* N-methylpyrrolidone, *THF* tetrahydrofuran.
[a]Reaction was conducted in a 10 mL autoclave.
[b]2.0 mmol of **3a** was added.
[c]10 mmol of **1a** and 2 mol% of Cp$_2$Ti(OTf)$_2$ were used.
[d]Isolated yield.

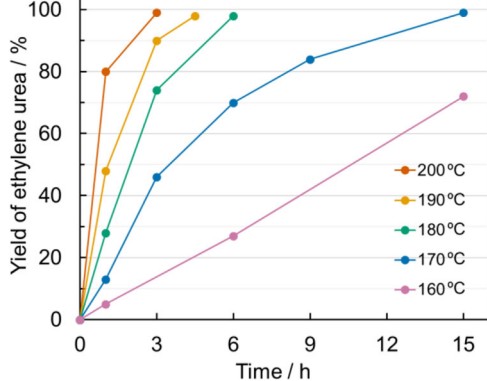

**Fig. 3 Effect of reaction temperature.** Plot of the NMR yield of ethylene urea as a function of reaction time at various temperatures: 200 °C (red curve), 190 °C (yellow curve), 180 °C (green curve), 170 °C (blue curve), and 160 °C (purple curve). Reaction conditions: **1a** (2.0 mmol), Ti (Cp)$_2$(OTf)$_2$ (0.20 mmol), and DMI (4.5 mL).

**Fig. 2 Reaction mechanism.** Plausible reaction mechanism for the Cp$_2$Ti (OTf)$_2$-catalyzed synthesis of ethylene urea **3a** from alkyl ammonium carbamates **2a**.

To determine the optimum reaction temperature, the reaction was conducted at various temperatures under the conditions corresponding to entry 3 in Table 2, and the yields at each temperature were plotted as a function of the reaction time (Fig. 3). The yield reached 99% within 15 h at temperatures >170 ° C. The reaction was first order with respect to the concentration of **1a**, and the activation energy calculated from the Arrhenius plot was 144 kJ/mol. Based on the above experimental results and kinetic analysis, 10 mmol of **1a**, 2 mol% of Cp$_2$Ti(OTf)$_2$ as a catalyst, and a temperature of 170 °C were determined to be the optimal reaction conditions. In addition, DMI was determined to be the best solvent as it is stable against acids and bases at high temperatures. No ethylene urea was formed when ethylenediamine was heated with 2 mol% of Cp$_2$Ti(OTf)$_2$ in DMI, confirming that DMI was not a carbonyl source. Under these conditions, **3a** was obtained in 99% NMR yield in 24 h, and

simple distillation following this allowed the isolation of **3a** in 82% yield (Table 2, entry 14).

Following the determination of the optimal reaction conditions, we investigated the substrate scope (Fig. 4). Initially, we investigated the synthesis and isolation of various alkyl ammonium carbamates from the corresponding amines and CO$_2$, which are commonly used in the conventional synthesis of urea derivatives, using the method for the synthesis of **1a** from ethylenediamine and CO$_2$ (refs. [24–29]). The reaction was conducted using EtOH or *n*-hexane as the solvent, depending on the solubility of the alkyl ammonium carbamates. Depending on the type of amine, intramolecular salts **1b**–**1g** or intermolecular salts **2a**–**2d** were obtained in 72–97% yields (see Supplementary Methods for further details). Elemental analysis of the solid alkyl ammonium carbamates, except **1b**, obtained by our method suggested that they were composed only of the corresponding amines and CO$_2$. The composition of **1b** was different from those of the other compounds probably due to the adsorption of small amounts of water and nitrogen gas. We shall

**Fig. 4 Substrate scope for the synthesis of urea derivatives.** Reaction conditions: alkyl ammonium carbamate (10 mmol), $Cp_2Ti(OTf)_2$ (0.04 mmol), DMI (4.5 mL), and 5 mL sealed reaction vessel. Values in parentheses indicate isolated yields. [a]Tentative isolated yield if pure **1b** was used. [b]8 mmol of **1c** and 4 mL of DMI were used. [c]2 mmol of alkyl ammonium carbamates and 10 mol% of $Cp_2Ti(OTf)_2$ were used for 15 h. [d]NMR yield could not be determined due to low solubility. [e]1,4-dioxane was used as the solvent instead of DMI for 24 h. [f]2 mmol of alkyl ammonium carbamate and 10 mol% of $Cp_2Ti(OTf)_2$ were used for 48 h. [g]4.6 mmol of **2d** and 4 mL of DMI were used.

assume that **1b** has an ideal composition and discuss it in the following sections, since **1b** is treated only as an intermediate raw material in this reaction. Urea derivative **3b** could also be synthesized in high yields when **1b**, with impurities such as water, was used. Compounds **3c** and **3d** with fused ring structures derived from **1c** and **1d** were synthesized, respectively, from the

cis- and trans-isomers of 1,2-cyclohexyl diamine **3c**, with relatively close distances between the two nitrogen atoms. While **3c** was obtained in a high yield (82%), **3d**, with relatively large distances between the two nitrogen atoms, was obtained in a relatively low yield (62%). When 10 mmol of **1d** was used, large amounts of insoluble white precipitate were formed. These were

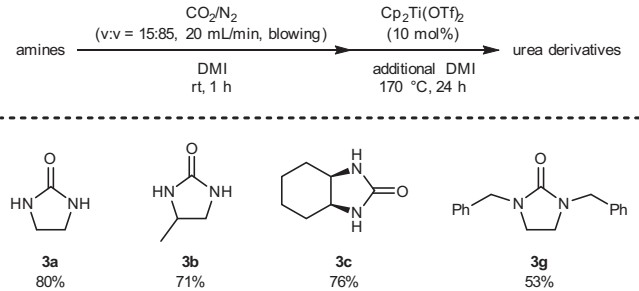

**Fig. 5 Substrate scope for the one-pot synthesis of urea derivatives.**
Reaction conditions: amines (2.0 mmol), $Cp_2Ti(OTf)_2$ (0.20 mmol), and DMI
(2.0 mL + 2.5 mL), and 5 mL sealed reaction vessel. Yields of urea derivatives
were determined by [1]H NMR, using mesitylene as an internal standard.

probably polyurea precipitates that were formed through an intermolecular reaction. Hence, it was necessary to lower the amount of **1d** to 2 mmol and increase the amount of catalyst to 10 mol% for the synthesis of **3d**. On the other hand, urea derivative **3e** was obtained in high yield (79%) even when **1e**, in which the carbon chain is longer than that in **1a–1d**, was used. Our method is also applicable to the synthesis of alkyl ammonium carbamates **1f** and **1g** derived from secondary diamines. The corresponding $N,N'$-disubstituted cyclic urea derivatives **3f** and **3g** can be obtained in relatively high isolated yields with 10 mol% of the catalyst (64% and 65%, respectively). Acyclic urea derivatives can also be synthesized from monoamine-derived alkyl ammonium carbamates, although the efficiency is slightly lower than that of the synthesis of cyclic urea derivatives. Alkyl ammonium carbamates **2a** and **2b** derived from linear primary amines gave corresponding urea derivatives **4a** and **4b** in moderate isolated yields (48% and 52%, respectively), when the reaction time was extended to 48 h, and 2 mmol of alkyl ammonium carbamates and 10 mol% of the catalyst were used. On the other hand, alkyl ammonium carbamate **2c** afforded corresponding urea derivative **4c** in low yield (13%), probably due to steric hindrance of the two cyclohexyl groups. Interestingly, the intermolecular alkyl ammonium carbamates **2d** derived from 2-aminobenzylamine gave $N$-aryl-$N'$-alkyl cyclic urea derivative **3h** in quantitative yield, unlike the case of monoamine-derived **2a–2c**. As mentioned above, our method can be applied to the synthesis of various alkyl ammonium carbamates, and all the urea derivatives can be easily isolated upon solvent removal and simple distillation. This method is particularly suitable for the synthesis of cyclic urea derivatives. In addition, even when alkyl ammonium carbamate **1a**, derived from $CO_2$ in ambient air or 15 vol% $CO_2$, was used as the raw material, urea derivative **3a** could be synthesized in the same high yield (99% NMR yield) as that obtained, using **1a** derived from pure $CO_2$.

In addition, we investigated the one-pot synthesis of urea derivatives from low concentrations of $CO_2$ (Fig. 5). To ethylenediamine (2.0 mmol) dissolved in DMI (2 mL; reaction solvent) in a 5 ml reaction vessel, 15 vol% $CO_2$ was blown at 20 mL/min for 1 h. Then, 10 mol% of $Cp_2Ti(OTf)_2$ and DMI (2.5 mL) were added, and the reaction vessel was sealed. After heating at 170 °C for 24 h, ethylene urea **3a** was obtained in 80% yield. This one-pot method can be applied to the synthesis of other urea derivatives. Using this one-pot method, urea derivatives **3b**, **3c**, and **3g**, which had particularly good yields in the stepwise synthesis, were obtained in relatively high yields (71%, 76%, and 53%, respectively).

## Conclusion

In this study, a method for the $Cp_2Ti(OTf)_2$-catalyzed synthesis of urea derivatives from alkyl ammonium carbamates, which can be

obtained from low concentrations of $CO_2$, including those in exhaust gases and ambient air, was developed. The various alkyl ammonium carbamates could be readily synthesized and isolated in high yields by exposing the corresponding amines in appropriate solvents to $CO_2/N_2$ (v:v = 15:85) mixed gases, in which the $CO_2$ concentration is almost equivalent to that in the exhaust gas of a thermal power plant. In addition, alkyl ammonium carbamate could be synthesized from ethylenediamine, using simulated exhaust gas and ambient air as a $CO_2$ source. Moreover, quality of the obtained alkyl ammonium carbamates was same, regardless of the $CO_2$ source. Our method was applicable to the synthesis of various other urea derivatives and particularly suitable for the synthesis of cyclic urea derivatives, such as ethylene urea, which has a high commercial value. Synergistic effect of the Ti catalyst, solvent (DMI), and the product (urea derivative) rendered the reaction highly efficient. Furthermore, we found that the urea derivatives could be synthesized from low concentrations of $CO_2$ and amines via alkyl ammonium carbamates in a one-pot manner, using DMI as a solvent.

This method for the synthesis of urea derivatives can be used to obtain useful chemicals from low concentrations of $CO_2$, without the need for expensive and energy-intensive processes of concentrating, compressing, and purifying $CO_2$. Moreover, since carbamate can be purified by bubbling low concentrations of $CO_2$ into the solvent, followed by filtration of the solid, this method has potential applications in utilizing exhaust gases containing various impurities. Therefore, we believe that our method for the synthesis of urea derivatives will aid in reducing $CO_2$ emissions by effectively using the $CO_2$ in ambient air and exhaust gases as a raw material.

## Methods

**General procedure**. For synthesizing urea derivatives, alkyl ammonium carbamates (10.0 mmol), $Cp_2Ti(OTf)_2$ (2 mol%), and DMI (4.5 mL) were added to a 5 mL autoclave reactor vessel and completely shielded with a stainless plate gasket. The vessel was dropped in a heated oil bath, and the reaction solution was stirred at 170 °C for 24 h. After the reaction reached completion, the vessel was removed from the oil bath and cooled to room temperature. The reaction mixture was extracted with MeOH (ca. 20 mL), and then 1,3,5-trimethylebenzene (ca. 100 mg) was added as an internal standard for [1]H NMR measurements to determine the NMR yield. After recording the [1]H NMR spectrum, MeOH was removed under reduced pressure using an evaporator. Then, components other than the target compound were roughly eliminated using a Kugelrohr apparatus. The obtained solid was washed with $Et_2O$/$n$-hexane and dried under reduced pressure at 50 °C to afford urea derivatives as a white solid. Full experimental details for all the alkyl ammonium carbamates and urea derivatives are provided in the Supplementary Methods and Supplementary Figs. 4–26.

## Data availability

Data supporting the findings of this study are available within this paper and its Supplementary Information are available from the corresponding author upon reasonable request.

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

## Acknowledgements

This work was supported by "Uncharted Territory Challenge 2050 (Mitou challenge 2050)" (K.T.) from New Energy and Industrial Technology Development Organization (NEDO)

## Author contributions

K.T., K.M., and J.-C.C. conceived the project and designed the experiments. K.T. and J.-C.C. directed the project. H.K. and K.T. performed the experiments and analyzed the results. H.K., K.T., K.M., and J.-C.C. wrote the manuscript. H.K., K.T., K.M., N.F., K.S., M.U., S.M., S.H., and J.-C.C discussed the results presented in this manuscript.

## Competing interests

The authors declare no competing interests.
