## [Peer Review File · Communications Chemistry]

Reviewers' comments:

Reviewer #1 (Remarks to the Author):

Choi and co-workers sent an interesting piece of work regarding the topic and some of the results. The authors report a new method for producing ureas from carbon dioxide in two steps. The first step, non-catalytic, involves the synthesis of the corresponding ammonium carbamate from carbon dioxide and an amine in ethanol at room temperature. The most interesting part of this simple method is that the authors show that just CO₂ from air or CO₂/N₂ mixtures with only 15% of CO₂ afford good results. In the second step the authors use the isolated carbamates to obtain ureas using a Ti complex as catalyst. The formation of carbamates from amines and CO₂ is described previously in the literature (ref 28). Moreover, many previous works (refs 23-26 and others not cited) reported the direct synthesis of ureas from amines and CO₂ without the isolation of the carbamate, as it is required in this protocol. Taking into account this, the novelty of the paper is based on the possibility of obtaining carbamates using diluted CO₂ and the optimization of a new catalyst (the Ti complex) for the urea synthesis. In my opinion, this can be not enough for a journal of general interest. However, I think the authors could perform additional efforts to compensate this and improve this work:

First, I suggest the authors to try the one pot system, I assume is very possible they already did and the results were not good, but I would encourage them to optimize it, maybe trying solvent mixtures or other catalysts for the second step, and of course I would like to see all these data in the article. I also think it would be very interesting that the authors determine the factors that explain why this system is so efficient capturing carbon dioxide at low pressures and so diluted. Furthermore, the authors only comment that they test group 4 metals for the second step, however in the SI (table s2) they show the use of several lewis acids (not only from group 4) and three bases, and there are really not so big differences in the results offered by some of them, even bases. I think a deeper study here is needed: to compare the reaction in table s2 conditions without catalyst, to compare with catalysts used in other papers (CeO₂, CsOH, organic bases) and to shed more light in the type of catalysis that is optimum in this reaction, maybe some other catalysts would be able to perform well in EtOH giving the one pot system at the mild conditions of the first step?

Other minor, but important, points to improve from the paper are:

- In page 3 when authors speak about direct methods to obtain ureas from CO₂ and amines I missed some cites: Green Chem., 2013, 15, 1567–1577 and ref 28.
- In page 5 the reference of Ingaki et al is wrong, should be ref 27
- In page 6, when authors say inversely proportional, should be directly
- In pages 8 and 9 (in the text) the authors wrote table 1 when should be table 2
- The ¹³C NMR of 2-ammonioethylecarbamate in page S7 show 4 signals, I think in this compound there are only 3 different carbons, could the authors explain this?
- In page 13 the authors comment the substrate scope, it would be more helpful for the reader that the examples where the catalyst loading should have been increased would be highlighted also in the text (not only in the footnotes).

Reviewer #2 (Remarks to the Author):

The authors reported organic urea synthesis from low concentration of CO₂ and diamines with a Cp₂Ti(OTf)₂ catalyst, and obtained high yields of the target ureas. Development of CO₂ fixation methods is a hot topic, and the development of effective catalyst systems is highly desirable. The important point of the catalyst system in this manuscript seems to be the addition effect of Cp₂Ti(OTf)₂ catalyst to the reaction, which brought high activity for the reaction, while the reaction can proceed without catalysts. Second important point is the acceleration effect of solvent and target urea on the reactivity. The phenomena are interesting. However, the authors did not discuss the effect of these factors and reaction mechanism. More detailed explanations of the reaction results, the effect of these factors and reaction mechanism will be required.

Other comments are shown below.

1. As the authors also know, ammonium carbamate is H₂NCOONH₄, and not for your case. Therefore, the authors should change the word "ammonium carbamate" to a suitable word, for example alkyl ammonium carbamate or carbamic acid alkylammonium salt.
2. In the introduction, the authors had better explain the reported complex catalyst systems effective for CO₂ transformation, particularly carbonate and carbamate syntheses, and explain the motivation of selection of the catalyst systems. Based on the previous your and other results, the authors can discuss the reaction mechanism.
3. From the NMR analyses (Fig. S1), the peak of free ethylene diamines was observed. This result indicates that the obtained white solid contains mixture of the carbamic acid salts and free ethylenediamine. The authors should calculate the purity of the carbamic acid salt by the NMR. It is very important information.
4. The authors showed only yields and did not show the conversion. Moreover, the pressure of the reactor is also important to understand the reaction system. The authors should show the information. Moreover, in some cases byproducts can be formed from the substrate and solvents. The authors had better show and discuss the results.
5. A large amount of catalyst was used for the reaction, but the catalyst amount should be reduced. The authors had better add the results of the catalyst amount effect on the reaction.
6. The authors selected DMI as a solvent, however the yield and reactivity in NMP solvent are similar to the case of DMI. NMP is a more common organic solvent compared with DMI. The authors should describe the reason.
7. The authors had better change the number of 96% to 93-96% in page 6.

Reviewer #3 (Remarks to the Author):

The paper reports on the synthesis of several urea derivatives by conversion of the relevant ammonium carbamates obtained by reaction of the corresponding amine/diamine and CO₂. All this chemistry is known since a long time (Green Chemistry 2010, 12, 1811; Green Chemistry, 2012, 14, 2899; see also Chapter 6 in ref. 11).

Pure CO₂, or CO₂/N₂ mixtures, or, even, atmospheric CO₂ have been used for the synthesis of ammonium carbamates. Based on the results obtained with the N₂/CO₂ (CO₂ 15% v/v) mixture the authors claim that the method "developed in this study should be effective in utilizing CO₂ of exhaust gases from industrial plants". This statement requires an experimental validation. Let me note that the presence of impurities such as H₂O, SO_x, NO_x in the gas mixture may cause the coprecipitation of ammonium carbonates, sulphates, nitrates that can affect the purity of the

ammonium carbamates to be used in the subsequent step of urea formation. The latter step is a catalytic step and the influence of the above impurities on the activity of the used catalyst should be evaluated. The clarification of this issue requires, therefore, further work involving, for example, the direct utilization, in the experiments, of samples of CO₂-containing exhaust gases from industrial sources.

The synthetic exploitation of atmospheric CO₂ is certainly of great interest. However, fixation of atmospheric CO₂ into ammonium carbamates under the used static conditions (see S.I.) is very slow (45% yield of 1a, after 1 week). This may be a serious drawback from the applicative point of view and makes very modest the contribution that the overall methodology can give in reducing the CO₂ concentration in the atmosphere, even if 1a is used to produce a bulk chemical such as ethylene-urea (see Introduction). The authors should explore different conditions (air bubbling, stirring, etc), albeit more energy intensive than the static conditions used in the present work. What happens if air is directly bubbled into pure 1a or into an ethanol solution of 1a? A kinetic study in this regard would be of interest. Moreover, is it really necessary to isolate ammonium carbamates before they are converted into the urea derivatives?

The best yields of 3a can be obtained in solvents such as NMP and DMI (Table 2). The efficient use of NMP in these reactions is well known (Green Chemistry 2012, 14, 2899). The authors have focussed their attention on the use of DMI as reaction solvent. The role of DMI in this process is not clear: it should be investigated in detail and fully elucidated. Is DMI involved in any reaction with 1a (or free ethylenediamine present at the equilibrium)? Is CO₂ initially fixed by diamine (into ammonium carbamate) selectively incorporated in the final product 3a? Can DMI act as CO source in the synthesis of 3a? Is there any evidence of DMI ring opening? The answer to the above questions can be of help to understand the autocatalytic effect of ethylene-urea under the conditions used in the present work (NB: the conditions used by the authors in the present work are very different from those reported in ref 28).

Additional comments:

Page 3: please provide some information about ethylene-urea global market.

page 6: "This suggested that the consumption rate of ethylenediamine was inversely proportional to the CO₂ concentration". Please check the above statement: according to Figure 1, the higher the concentration of CO₂, the faster the conversion of ethylenediamine.

Fig.s S1 and S2: please provide the assignments of the signals. Did you observe any decomposition of the zwitterionic carbamate in D₂O?

page S8: "Determination of conversion rate for 1,2-diaminoethane during CO₂ bubbling." For the sake of clarity, please provide in S.I. a copy of a proton spectrum (in dmsd-d₆) used for the determination of ethylenediamine conversion.

In summary, I think that this paper requires further experimental work and a very major revision addressing the above issues before publication in the Journal.

Response to Reviewer #1:

We thank you for reviewing our manuscript. We value your comments, and we have presented below the point-by-point responses to your comments.

(Remarks to the Author):

Choi and co-workers sent an interesting piece of work regarding the topic and some of the results. The authors report a new method for producing ureas from carbon dioxide in two steps. The first step, non-catalytic, involves the synthesis of the corresponding ammonium carbamate from carbon dioxide and an amine in ethanol at room temperature. The most interesting part of this simple method is that the authors show that just CO₂ from air or CO₂/N₂ mixtures with only 15% of CO₂ afford good results. In the second step the authors use the isolated carbamates to obtain ureas using a Ti complex as catalyst. The formation of carbamates from amines and CO₂ is described previously in the literature (ref 28). Moreover, many previous works (refs 23-26 and others not cited) reported the direct synthesis of ureas from amines and CO₂ without the isolation of the carbamate, as it is required in this protocol. Taking into account this, the novelty of the paper is based on the possibility of obtaining carbamates using diluted CO₂ and the optimization of a new catalyst (the Ti complex) for the urea synthesis. In my opinion, this can be not enough for a journal of general interest. However, I think the authors could perform additional efforts to compensate this and improve this work:

First, I suggest the authors to try the one pot system, I assume is very possible they already did and the results were not good, but I would encourage them to optimize it, maybe trying solvent mixtures or other catalysts for the second step, and of course I would like to see all these data in the article. I also think it would be very interesting that the authors determine the factors that explain why this system is so efficient capturing carbon dioxide at low pressures and so diluted. Furthermore, the authors only comment that they test group 4 metals for the second step, however in the SI (table s2) they show the use of several lewis acids (not only from group 4) and three bases, and there are really not so big differences in the results offered by some of them, even bases. I think a deeper study here is needed: to compare the reaction in table s2 conditions without catalyst, to compare with catalysts used in other papers (CeO₂, CsOH, organic bases) and to shed more light in the type of catalysis that is optimum in this reaction, maybe some other catalysts would be able to perform well in EtOH giving the one pot system at the mild conditions of the first step?

Response: We thank you for explaining the need to develop the one-pot synthesis and also for proposing the experimental method for the same. After thorough investigations, we found that alkyl ammonium carbamate could be generated from low concentrations of CO₂ in DMI and could be directly converted to urea derivatives in high yield in a one-pot manner. This one-pot synthesis method has added to the manuscript. We had also tried using CeO₂, CsOH, and other organic bases as a catalyst; however, they did not show high activity. These results were also added to the SI.

Other minor, but important, points to improve from the paper are:

1. In page 3 when authors speak about direct methods to obtain ureas from CO₂ and amines I missed some cites: Green Chem., 2013, 15, 1567–1577 and ref 28.

Response: We thank you for your comment. The suggested papers have been cited and added to the references as per your suggestion.

2. In page 5 the reference of Ingaki et al is wrong, should be ref 27

Response: We thank you for your comment. In Page 6, the reference was corrected as per your suggestion.

3. In page 6, when authors say inversely proportional, should be directly

Response: We thank you for your suggestion. In Page 6, “This suggested that the consumption rate of ethylenediamine was inversely proportional to the CO₂ concentration” has been revised as “In other words, although the rate of carbamate formation depended on the CO₂ concentration, the carbamate could be obtained in high yields even with a low concentration of CO₂ if the reaction time was prolonged” in the text.

4. In pages 8 and 9 (in the text) the authors wrote table 1 when should be table 2

Response: We thank you for your highlighting this. This has now been corrected.

5. The ¹³C NMR of 2-ammonioethylcarbamate in page S7 show 4 signals, I think in this compound there are only 3 different carbons, could the authors explain this?

Response: We thank you for your comment. Since 2-ammonioethylcarbamate **1a** is insoluble in organic solvents, NMR can only be measured in D₂O. However, if **1a** is dissolved in D₂O, some amount of the carbamate is probably converted to the salt of carbonate and proton-bridged ammonium (structures shown below). Therefore, in addition to the three signals of 2-ammonioethylcarbamate, very weak signals corresponding to quaternary carbon of the carbonate ion (which is also slightly visible in the spectrum in page S7) and signals corresponding to proton-bridged ammonium are observed. Thus, elemental analysis was used to confirm that solid 2-ammonioethylcarbamate contained almost no salt of carbonate and proton bridged ammonium before dissolution in D₂O. This has been added to the main text and SI.

6. In page 13 the authors comment the substrate scope, it would be more helpful for the reader that the examples where the catalyst loading should have been increased would be highlighted also in the text (not only in the footnotes).

Response: We thank you for your comment. The following was added in Page 16:

“Hence, it was necessary to lower the amount of **1d** to 2 mmol and increase the amount of catalyst to 10 mol% for the synthesis of **3d**.”

Response to Reviewer #2:

We thank you for reviewing our manuscript. We value your comments, and we present below the point-by-point responses to your comments:

(Remarks to the Author)

The authors reported organic urea synthesis from low concentration of CO₂ and diamines with a Cp₂Ti(OTf)₂ catalyst, and obtained high yields of the target ureas. Development of CO₂ fixation methods is a hot topic, and the development of effective catalyst systems is highly desirable. The important point of the catalyst system in this manuscript seems to be the addition effect of Cp₂Ti(OTf)₂ catalyst to the reaction, which brought high activity for the reaction, while the reaction can proceed without catalysts. Second important point is the acceleration effect of solvent and target urea on the reactivity. The phenomena are interesting. However, the authors did not discuss the effect of these factors and reaction mechanism. More detailed explanations of the reaction results, the effect of these factors and reaction mechanism will be required.

Response: We thank you for highlighting the aspect of reaction mechanism. A description of the reaction mechanism was added, referring to the reaction mechanism and the autocatalytic effect described in previous reports on the synthesis of urea derivatives from amines and CO₂. In fact, we have cited the reaction mechanism of the CeO₂ acid-base functional solid catalyst proposed by Tomishige et al. We believe that Cp₂Ti(OTf)₂ acts as a Lewis acid, and the solvent, such as DMI or NMP, acts as a Lewis base.

Other comments are shown below.

1. As the authors also know, ammonium carbamate is H₂NCOONH₄, and not for your case. Therefore, the authors should change the word “ammonium carbamate” to a suitable word, for example alkyl ammonium carbamate or carbamic acid alkylammonium salt.

Response: As suggested, we have now changed “ammonium carbamate” to “alkyl ammonium carbamate,” as the former is misleading.

2. In the introduction, the authors had better explain the reported complex catalyst systems effective for CO₂ transformation, particularly carbonate and carbamate syntheses, and explain the motivation of selection of the catalyst systems. Based on the previous your and other results, the authors can discuss the reaction mechanism.

Response: As suggested, in Pages 9 and 10, we have now added the description of the catalytic synthesis of carbonates and carbamates from CO₂ and alcohols or amines (before the section of catalyst screening) to explain the basis for the catalyst selection.

3. From the NMR analyses (Fig. S1), the peak of free ethylene diamines was observed. This result indicates that the obtained white solid contains mixture of the carbamic acid salts and free ethylenediamine. The authors should calculate the purity of the carbamic acid salt by the NMR. It is very important information.

Response: We thank you for your comment. In Fig S1, the signal other than the signal for **1a** is not from the unreacted ethylenediamine but from the bridging ammonium salts formed upon the reaction of **1a** with the D₂O solvent. Since alkyl ammonium carbamates such as **1a** are soluble only in water, it is difficult to avoid the appearance of this signal. Therefore, we considered that elemental analysis would be the best technique to determine the composition and purity. Elemental analysis confirmed that all the synthesized carbamate solids contained almost no impurities (details in the SI).

4. The authors showed only yields and did not show the conversion. Moreover, the pressure of the reactor is also important to understand the reaction system. The authors should show the information. Moreover, in some cases byproducts can be formed from the substrate and solvents. The authors had better show and discuss the results.

Response: We thank you for your comment. In this reaction, alkyl ammonium carbamates were completely consumed, giving only the corresponding urea derivatives and amines. Thus "conversions" are omitted (i.e., the conversion was 100% for all the reactions). In addition, because the reaction temperature is lower than the boiling point of DMI, the reaction pressure was not so high. Nevertheless, we have now added the results (in the SI) of the system with the pressure gauge connected. It is to be noted that when a pressure gauge is connected, the yield decreases due to the head space.

5. A large amount of catalyst was used for the reaction, but the catalyst amount should be reduced. The authors had better add the results of the catalyst amount effect on the reaction.

Response: We thank you for your comment. The amount of catalyst was reduced to 2 mol% (Table 2-Entry 14). In addition, when the reaction was carried out at 1 mol% of catalyst, the yield was apparently decreased. Therefore, we concluded that 2 mol% of catalyst was the best amount to complete the reaction within 24 h.

6. The authors selected DMI as a solvent, however the yield and reactivity in NMP solvent are similar to the case of DMI. NMP is a more common organic solvent compared with DMI. The authors should describe the reason.

Response: We thank you for your comment. In Page 12, we have now explained that DMI was determined to be the best solvent because it is stable against acids and bases at high temperatures compared to NMP and other solvents.

7. The authors had better change the number of 96% to 93-96% in page 6.

Response: We thank you for your comment. As per your suggestion, “96%” was changed to “93–96%”.

Response to Reviewer #3:

We thank you for reviewing our manuscript. We value your comments, and we present below the point-by-point responses to your comments.

(Remarks to the Author)

The paper reports on the synthesis of several urea derivatives by conversion of the relevant ammonium carbamates obtained by reaction of the corresponding amine/diamine and CO₂. All this chemistry is known since a long time (Green Chemistry 2010, 12, 1811; Green Chemistry, 2012, 14, 2899; see also Chapter 6 in ref. 11).

Pure CO₂, or CO₂/N₂ mixtures, or, even, atmospheric CO₂ have been used for the synthesis of ammonium carbamates. Based on the results obtained with the N₂/CO₂ (CO₂ 15% v/v) mixture the authors claim that the method “developed in this study should be effective in utilizing CO₂ of exhaust gases from industrial plants”. This statement requires an experimental validation. Let me note that the presence of impurities such as H₂O, SO_x, NO_x in the gas mixture may cause the coprecipitation of ammonium carbonates, sulphates, nitrates that can affect the purity of the ammonium carbamates to be used in the subsequent step of urea formation. The latter step is a catalytic step and the influence of the above impurities on the activity of the used catalyst should be evaluated. The clarification of this issue requires, therefore, further work involving, for example, the direct utilization, in the experiments, of samples of CO₂-containing exhaust gases from industrial sources.

Response: We thank you for the suggestion on using exhaust gas. We investigated the synthesis of alkyl ammonium carbamate **1a** using simulated exhaust gas (CO₂: 15 vol%, N₂: 85 vol%, CO: 300 ppm, NO₂: 500 ppm, SO₂: 500 ppm); the composition was determined based on the details of exhaust gas in the literature. Although the simulated exhaust gas was difficult to add water as steam, we did not consider it because the EtOH we used was not dehydrated, and there was originally a larger amount of water than is expected to dissolve from the bubbling exhaust gas volume. As a result, **1a** was obtained with purity and yield that were comparable to those obtained using 15 vol% CO₂ (purity was confirmed by elemental analysis). The subsequent synthesis of ethylene urea proceeded without any difficulty. This is because of the highly active impurities such as CO, NO₂, and SO₂ in the simulated exhaust gas were only present in trace amounts, in the order of ppm, and the flow volume required for the quantitative synthesis of **1a** had a negligible effect on the reaction.

The synthetic exploitation of atmospheric CO₂ is certainly of great interest. However, fixation of atmospheric CO₂ into ammonium carbamates under the used static conditions (see S.I.) is very slow (45% yield of **1a**, after 1 week). This may be a serious drawback from the applicative point of view and makes very modest the contribution that the overall methodology can give in reducing the CO₂ concentration in the atmosphere, even if **1a** is used to produce a bulk chemical such as ethylene-urea (see Introduction). The authors should explore different conditions (air bubbling, stirring, etc), albeit more energy intensive than the static conditions used in the present work. What happens if air is directly bubbled into pure **1a** or into an ethanol solution of **1a**? A kinetic study in this regard would be of interest. Moreover, is it really necessary to isolate ammonium carbamates before they are converted into the urea derivatives?

Response: We thank you for your comment. Bubbling and stirring have also been examined, and it was found that the volatilization of ethylenediamine was predominant, resulting in a significant decrease in the yield of alkyl ammonium carbamate. In addition, when an ethanol solution of ethylenediamine was used, the alkyl ammonium carbamates could not be isolated due to the contamination of water from air. Moreover, we believe that quiescent condition is the most energy-efficient condition, as it requires no energy even though the reaction takes a long time. However, if a higher concentration of CO₂ is used, such as that in exhaust gas, it would be optimal to use it. Furthermore, as mentioned in the response to reviewer #1, we found that urea derivatives could be synthesized from 15% CO₂ in relatively high yields in a one-pot manner using DMI as the solvent. We have now added this information in the revised manuscript. However, considering the use of exhaust gas and air, we believe that the stepwise method is also worthwhile.

The best yields of 3a can be obtained in solvents such as NMP and DMI (Table 2). The efficient use of NMP in these reactions is well known (Green Chemistry 2012, 14, 2899). The authors have focussed their attention on the use of DMI as reaction solvent. The role of DMI in this process is not clear: it should be investigated in detail and fully elucidated. Is DMI involved in any reaction with 1a (or free ethylenediamine present at the equilibrium)? Is CO₂ initially fixed by diamine (into ammonium carbamate) selectively incorporated in the final product 3a? Can DMI act as CO source in the synthesis of 3a? Is there any evidence of DMI ring opening?

The answer to the above questions can be of help to understand the autocatalytic effect of ethylene-urea under the conditions used in the present work (NB: the conditions used by the authors in the present work are very different from those reported in ref 28).

Response: We thank you for your comment. Since there is no formation of DMI-derived amine in the reaction mixture after the completion of reaction, DMI was not considered to be a CO source. In addition, since ethylene urea is quantitatively produced even with NMP, it is reasonable to assume that DMI is not necessarily essential and that the entire CO₂ in the system is converted to ethylene urea. Furthermore, as mentioned in the response to reviewer #2, we have now added the details of the reaction mechanism in the revised manuscript. By revisiting the mechanism, we proposed that the reaction proceeds efficiently by a concerted acid-base reaction, in which the titanium catalyst acts as an acid and the solvent, such as DMI, acts as a base.

Additional comments:

Page 3: please provide some information about ethylene-urea global market.

Response: We thank you for your suggestion. In Page 3, we have now added the following information:

“In particular, ethylene urea, a cyclic urea derivative, is a relatively high value-added core chemical (ca. 10 USD/kg in 2018) with a large market (ca. 12,000 t/year in 2018). It is used in paints and as a fabric finishing agent and raw material for agrochemicals.”

page 6: “This suggested that the consumption rate of ethylenediamine was inversely proportional to the CO₂ concentration”. Please check the above statement: according to Figure 1, the higher the concentration of CO₂, the faster the conversion of ethylenediamine.

Response: We apologize for this oversight. In Page 6, “This suggested that the consumption rate of ethylenediamine was inversely proportional to the CO₂ concentration” has now been changed to “In other words, although the rate of carbamate formation depended on the CO₂ concentration, the carbamate could be obtained in high yields even with a low concentration of CO₂ if the reaction time was prolonged.”

Fig.s S1 and S2: please provide the assignments of the signals. Did you observe any decomposition of the zwitterionic carbamate in D₂O?

Response: We thank you for highlighting this. We have now added the NMR assignments. As mentioned in the responses to reviewers #1 and #2, some of the signals originated from the bridging ammonium salt produced by the reaction of **1a** with D₂O. However, the composition and purity of the alkyl ammonium carbamate were confirmed by elemental analysis.

page S8: “Determination of conversion rate for 1,2-diaminoethane during CO₂ bubbling.” For the sake of clarity, please provide in S.I. a copy of a proton spectrum (in dmsO-d₆) used for the determination of ethylenediamine conversion.

Response: We thank you for your suggestion. We have now added the corresponding ¹H NMR spectra in the SI.

REVIEWERS' COMMENTS:

Reviewer #1 (Remarks to the Author):

Takeuchi, Choi and co-workers made an important effort to address all the points and the quality of the manuscript was clearly improved. In my opinion, the work is suitable to be published.

Reviewer #2 (Remarks to the Author):

The authors added explanations and revised the manuscript properly according to the comments. Therefore, the manuscript is ready for publication in Communications Chemistry.

Reviewer #3 (Remarks to the Author):

The paper has been suitably revised. I recommend publication in the Journal after addressing the following minor issue :

Page 5, line 3 bottom and caption of Fig. 1: in the revised version the value of the gas flow rate has been modified ("50mL/min" in place of the original value of "100 mL/min"). However, in the Supplementary Material (page S10) the reported value for the used gas flow rate is "100 mL/min". Which is the correct value?

Dear Dr. Ortner:

Thank you very much for spending your time to evaluate our manuscript. (COMMSCHEM-20-0392A). Based on your information, reviewers #1 and #2 would like to accept our manuscript. We have read the comments from reviewer #3. Please check our response below.

Thank you for your consideration. I look forward to hearing from you.

Sincerely,

Jun-Chul Choi

Reviewer #1 (Remarks to the Author):

Takeuchi, Choi and co-workers made an important effort to address all the points and the quality of the manuscript was clearly improved. In my opinion, the work is suitable to be published.

Reviewer #2 (Remarks to the Author):

The authors added explanations and revised the manuscript properly according to the comments. Therefore, the manuscript is ready for publication in Communications Chemistry.

Reviewer #3 (Remarks to the Author):

The paper has been suitably revised. I recommend publication in the Journal after addressing the following minor issue:

Page 5, line 3 bottom and caption of Fig. 1: in the revised version the value of the gas flow rate has been modified ("50mL/min" in place of the original value of "100 mL/min"). However, in the Supplementary Material (page S10) the reported value for the used gas flow rate is "100 mL/min". Which is the correct value?

Response: We thank you for your suggestion. "50mL/min" is the correct value, so the flow rate value in SI has been corrected.